# Social Capital Inequality According to Hukou in Unequal Economic Environments in China

**Songyang Lyu, Sungik Kang ***  **and Ja-Hoon Koo ***

Department of Urban and Regional Development, Hanyang University, Seoul 04763, Korea
* Correspondence: namugnel@gmail.com (S.K.); jhkoo@hanyang.ac.kr (J.-H.K.); Tel.: +82-2-2220-0339 (J.-H.K.)

**Abstract:** China is experiencing an increase in socioeconomic inequality in comparison to the global trend. Employing the hukou registration as a focal point, this study seeks to examine social capital differences between developed and underdeveloped regions in China. As the data for the analysis were from the China Family Panel Studies, social capital was measured by social trust, participation, and networks. The Gini coefficient, a measure of economic inequality, was calculated for 25 different provinces in China in this paper. In light of the fact that these are panel data collected between 2014 and 2018, this study employed the random-effect model for panel analysis. The first finding is that individuals, in an environment characterized by high levels of economic inequality, have low levels of social capital levels in China. Second, the inverse relationship between economic inequality and social capital varies according to social capital element. Specifically, this inverse relationship was observed in social trust and social networks, but not in social participation. Third, hukou registration moderated the inverse relationship between economic inequality and social capital. The rural hukou registration revealed a stronger inverse relationship between economic inequality and social capital than its urban counterpart. This indicates that the negative impact of economic inequality on social capital resulted in additional inequality among rural hukou holders.

**Keywords:** economic inequality; social capital; socioeconomic status; hukou; China inequality

## 1. Introduction

Globally, socioeconomic inequality is increasing, and rising economic disparities threaten social stability [1]. The wealth gap between the top 20% and the remaining 80% has become more pronounced in recent years, and the income growth rate of the top 20% is particularly steep [2]. The negative impact of socioeconomic inequality is the segregation of residence according to wealth [3,4], and inequality is negatively associated with problems, such as social insecurity, anxiety, disease, and increased crime [5]. In addition, increasing inequality contributes to the deterioration of communities and lowers the quality of society [1,2,6].

China's GDP has climbed at a rate of 9.7 percent per annum since the reform and open-door policy was introduced in the late 1970s, and the real GDP per capita soared to a factor of nine by 2007 [7]. According to data published by the US Census Bureau, China's Gini coefficient was approximately 0.30 in 1980, but had almost quadrupled to 0.55 by 2012, which is considerably above the United States' level of 0.45 [8]. Moreover, in recent years, China's economic disparity has far surpassed the average, despite the fact that other nations with comparable economic growth have greater Gini coefficients [8]. In the dual economy with urban-rural segmentation, urban inhabitants usually have a larger family income level than rural inhabitants [9]. The economic polarization between urban and rural areas has become prominent, which has attracted extensive attention in China. Economic inequality between urban and rural residents has become one of the most important types of economic inequality in China [10].

One of the adverse effects of socioeconomic inequality in our society is a reduction in social capital. According to a study by Putnam (2000) [11], social capital declined, along

with increasing economic inequality throughout the 20th century. Wilkinson's (2005) [1] analysis explains that the widening of the economic disparity is closely related to the rise of individualism and the decline in social capital. In addition, the middle and upper classes tend to be neighbors with similar income groups, and regional wealth segregation becomes more severe [12]. The segregation of residences according to wealth disparity leads to a social separation between schools, churches, and local communities, and negatively affects social capital formation [6].

In an unequal society, the importance of social status is further emphasized. An unequal society is one in which the disparity related to social status is larger, and competition for status is more intense than that in an equal society [13]. The hukou registration system is a basic institutional arrangement in Chinese society, and it is also a system centered on hukou registration and management [14]. The hukou system takes hukou as an important credential for resource allocation and benefit distribution, which has a great impact on social stratification and mobility [14,15]. The core content of the hukou registration system includes a dual identity system that divides citizens into urban and rural registrations. At the same time, according to the principle of hukou jurisdiction, strict administrative control is exercised on the migration of hukou between different places. This institutional arrangement has had an important impact on the formation of the urban–rural dual structure of Chinese society and the emergence of urban hierarchies through the control of identity conversion and autonomous migration [14]. In China, a representative of social status is a hukou registration. Hukou registers are divided into urban and rural areas according to their original birth regions, and access and resources for various social benefits vary according to hukou registration [16,17]. In other words, social inequality related to income, education, and freedom of movement occurs according to the hukou registration. In particular, there are various social restrictions as rural hukou registers live in urban areas [16,18]. In addition, studies on social capital in the West have emphasized that socioeconomic status (SES) is a significant factor in creating inequality in social capital formation [19–21]. Thus, China's representative SES is the hukou register system, and it is necessary to study whether this system is associated with social capital gap in Chinese society.

Consequently, China is also transitioning into an unequal society, which can have a negative impact on the accumulation of social capital. In addition, China's unequal environment is a society in which the significance of hukou registration is further emphasized, and social inequality may result from hukou registration. In an unequal environment, the level of social capital differs based on hukou registers, according to a meta-analysis of existing studies. However, few studies have investigated the role of hukou registration in the relationship between inequality and social capital. This study examines the mechanisms of income inequality and social capital of China's residents, as well as the moderation role of hukou, to comprehend the relationship between economic inequality and social capital in developed and deprived regions of China. In conclusion, the findings of this paper remain robust after we tested for the endogeneity problem.

## 2. Literature Review

### 2.1. Social Capital in Unequal Region

As the economy grows focused on several areas within the city, hierarchies and socioeconomic classes within the city arise. Due to urban globalization, economic inequality occurs in cities as technology and human resources are concentrated in specific regions [6,22]. In addition, the international population movement, due to the development of transportation, has accelerated inequality [23], and the spread of automobiles in American society has divided the space between affluent suburban areas and poor urban centers [6]. The concentration of human capital and the occurrence of socioeconomic inequality result in spatial segregation according to the wealth class within the city [24]. As a result, the benefits of urban innovation and economic growth, due to the concentration of human resources and economic resources, created a gap between the upper and lower classes who already enjoyed the benefits [6].

It has been hypothesized that an increase in economic disparity leads to a decrease in the social interaction between different classes, and consequently, a decline in social capital [1,25]. Social capital is a term that describes the level of social networks, reciprocity, trust, and norms of individuals. Social capital has attracted attention because society exerts strong civic power when it has high social participation and networks [11]. As a concept with various meanings, social capital is divided into perceived elements, such as trust and reciprocity, and physical elements, such as participation and cooperation [11,26]. As a result, research on social capital must be divided into structural and cognitive elements.

An increase in economic inequality has a strong correlation with a decrease in social quality, such as a decline in social trust, a reduction in community participation, an increase in hostility, and a drop in the social network. Eric (2013) [27] analyzed the impact of income level and income inequality on social trust levels in 20 countries using the World Income Inequality Database. The results of this study highlighted that income level alone was insufficient to explain social trust and that income inequality further reduced the decline in social trust by approximately 13%. A study by Putnam (1993) [28] examined the level of voluntary participation in local groups and participation in community life in 20 Italian provincial governments. The results of this study demonstrated that the income inequality index and the community participation indices showed a very close relationship.

Putnam's study, which analyzed the correlation between income distribution inequality and the social capital index in 50 US states, pointed out that the change in social capital and income disparity was almost precisely the same (Putnam, 2000) [11]. A study analyzing the association between the Robin Hood Index and social capital factors in 39 states in the United States illustrated that income inequality was closely related to group participation and the social capital index [25]. Wilkinson (2005) found that the deepening of income inequality has a structural mechanism in which the social distance between income groups increases and the quality of social relationships deteriorates. He described that the deepening of income inequality has worsened the quality of social relationships, such as weakening trust levels and lower participation in community life, by widening social status disparities and increasing social distance between wealth groups [1].

## 2.2. Research Assumption Based on Hukou

Hukou registration is an essential social factor for studying inequality in China. The Hukou register is one of the representative social inequality factors in China, and access and benefits to social resources, such as education, health care, and welfare entitlements, differ depending on the family register [29]. Hukou registration is divided into urban and rural registers, and the government created a registration system to establish urban and rural areas [30]. People with rural hukou are generally more disadvantaged socially and fewer opportunities than people with urban hukou [16]. Families who immigrated to cities with hukou registers from rural areas have relatively greater economic resources and greater opportunity restrictions than those from urban areas [17]. For example, some public schools in cities do not allow children from rural hukou to attend, leading to educational inequality [31]. According to a study by Afridi et al. (2015) [16], rural hukou have lower earnings performance in the piece rate regime than urban hukou. In other words, China's social system is fundamentally linked to social inequality through its hukou registration.

## 2.3. Contribution

Based on this research background, the objectives of this study are as follows:

**H1.** *Similar to developed countries, China also has low social capital in an environment of high economic inequality.*

**H2.** *The inverse relationship between inequality and social capital varies by social capital element.*

**H3.** *The inverse relationship between inequality and social capital is more detrimental for individuals for rural hukou holders.*

This study aimed to understand the relationship between inequality and social capital in China through hukou registration. In other words, we investigate whether social inequality arising from hukou registration is present in the negative relationship between economic inequality and social capital. We begin with the premise that, even in China, the level of individual social capital is low due to the prevalence of inequality. We attempt to demonstrate the connection between economic inequality and social capital on the basis of this assumption. The second assumption is that the negative relationship between economic inequality and social capital varies by social capital element. The empirical models classified the elements of social capital as social trust, participation, and network. The third assumption is that the inverse relationship between economic inequality and social capital would be more detrimental for individuals that are rural hukou holders. Correspondingly, the moderating effect of hukou registration on the relationship between economic inequality and social capital was analyzed.

## 3. Materials and Methods

### 3.1. Data

This study used the China Family Panel Studies (CFPS) data for social capital and covariate variables. This data set is a biennial follow-up survey that aims to reflect China's economic development and social changes through follow-up surveys of sample villages, households, and family members nationwide. Data were collected and released by the China Social Science Survey Center of Peking University and the Survey and Research Center of the University of Michigan, USA. Five major questionnaires were designed in the CFPS: The community questionnaire, the family roster questionnaire, the family questionnaire, the child questionnaire, and the adult questionnaire. Eligible individuals were surveyed at the individual level. In this study, we targeted Chinese citizens aged 18 years or older. In the 2010 baseline survey, the subjects of the survey were 25 provinces and cities across the country, 161 districts and counties, and 649 villages, with a total of 15,000 households and 57,155 individuals. We used sample data to cover 25 provinces, municipalities, and autonomous regions in 2014, 2016, and 2018 (except Hong Kong, Macau, Taiwan, Xinjiang, Tibet, Qinghai, Inner Mongolia, Ningxia, and Hainan) and those encompassed by the CFPS account for nearly 95% of all Chinese residents living in mainland China. After deleting observations with missing or incomplete variable information, the final sample size for this study was 40,770 people.

### 3.2. Variables

3.2.1. Social Capital Variable

Social capital is a broad theoretical construct [32] and it describes the resources and advantages we acquired via our relationships with others, either as people or as organizations [33]. Social capital can be further delineated as structural and cognitive social capital [33]. Previous studies have measured that trust is especially vital to the success of large organizations [34]. Therefore, in this study, we used social trust and trust in the neighborhood as variables for the cognitive dimension. We focused on the question of trust in neighbors in the questionnaire, that is, "How much do you trust your neighbors?". A total of 0–10 is an option, and a higher score indicates more trust. Social capital also occured mainly through social participation, cooperation with neighbors, and so on [35]. With the rapid development of China's economy and society, resident participation has continuously improved. Social participation includes donating to charitable organizations in society and participating in voluntary organizations [36]. Donating, or charitable giving, is the active transfer of money or cash equivalent to society or the community. Donations are performed to benefit others beyond their own family [37]. Therefore, we used social participation to reflect the state of structural social capital. In this paper, for social participation, we

used the question of social donation expenditure (yuan/year) in the household economic questionnaire, that is, "in the past 12 months, how much social donation did your family make in cash and in-kind". The majority of people saw the personalist concept of 'particularistic relationship structure' as the foundation of Confucian principles, which controlled Chinese culture [38]. To a greater extent than in other cultures, social networks and guanxi are more central to the concept of family social capital in China [39]. The representative word in China is 'Guanxi,' which means the action of 'gifts' in order to form and maintain a relationship in China [40]. The exchange of monetary gifts has long been fundamental to social interactions in China. People who give and receive gifts are not only relatives, but also people around them [41]. Therefore, we used social networks and gifts as a structural social capital. We used the question, "In the past 12 months, what was the total amount of money your family received in gifts and cash?". These factors constitute the most popular forms of social capital and have been repeatedly peer-reviewed and cross-validated.

### 3.2.2. Economic Inequality Index and Hukou

There are also many measures of economic inequality, among which the most widely used is the Gini index [42]. In developing countries, individual income is not accurately measured because many economic activities do not occur in an informal exchange market [43]. Moreover, income cannot fully reflect residents' living standards and economic levels, especially for families with relatively scarce resources [44]. When measuring the gap between the rich and the poor, consumption inequality is an adequate performance and persuasive index for income inequality because it represents individual and household economic levels [45]. Thus, we mainly used the economic consumption Gini index from the CFPS data at the provincial level as the core independent variable. The Gini coefficient is a figure between 1 and 0, where 1 indicates perfect inequality and 0 indicates perfect equality. In this study, zero economic inequality represented areas with similar household consumption and a more egalitarian economic environment. In contrast, 1 represents inequality regions with large disparities in household consumption. The economic inequality Gini index of China reached its lowest at 0.329 in 2014 and its highest at 0.557 in 2016.

Each citizen is required to legally register at a household police station from birth, and this registration is known as personal identification [46]. The household registration records the type of hukou type, legal address, up to affiliation, and the other personal and family details of Chinese citizens [47]. In this paper, we used the question, "current household registration type is?". The type of "Agricultural" is rural hukou, and the type of "Non-Agricultural" is urban hukou. Thus, in this paper, hukou, a dummy variable, is coded as rural hukou = 0 and urban hukou = 1.

### 3.2.3. Covariate Variable

This research considered gender, age, education, homeownership, and religion as demographic and socioeconomic covariate variables combined with social capital [11,26,48]. Using multiple graphs, Putnam (2000) demonstrated that the perception and degree of social capital varied by age and gender. Community involvement is closely connected with educational success [49]. Regional and individual economic variations produce disparities in community ties owing to unequal investment in social capital-related programs and services [50]. As such, the household income, employment, and house type are individual and regional economic factors. Families have characteristics, such as high-density networks and collective support, and the family makeup is connected to the social and interdependent relationships of individuals [51]. Therefore, family composition and marital status were added as covariate factors. Homeownership is a characteristic associated with local settlement, and the area with a significant homeownership rate serves as the foundation for social capital data [52]. Finally, the duration of the interaction with neighbors is a social capital element [52,53]. Table 1 displays the descriptive statistics and measurement information for this investigation in this study.

**Table 1.** Variable list and descriptive statistics.

| Variables | Observation | Mean | S.D. | Min | Max |
|---|---|---|---|---|---|
| Dependent variable | | | | | |
| Social trust | 40,770 | 6.701 | 2.128 | 0.000 | 10.000 |
| Social participation | 40,770 | 1.030 | 2.167 | 0.000 | 10.820 |
| Social network | 40,770 | 7.605 | 1.840 | 0.000 | 12.707 |
| Independent variable | 40,770 | | | | |
| Economic inequality index | 40,770 | 0.461 | 0.032 | 0.329 | 0.557 |
| Hukou (1 = urban, 0 = rural) | 40,770 | 0.278 | 0.448 | 0.000 | 1.000 |
| Social demographic characteristic | 40,770 | | | | |
| Age (actual age, value ranges: 18–90) | 40,770 | 48.494 | 14.530 | 18 | 90 |
| Apartment (1 = yes, 0 = no) | 40,770 | 0.186 | 0.389 | 0.000 | 1.000 |
| Education (above seminar high school = 1, under = 0) | 40,770 | 0.245 | 0.430 | 0.000 | 1.000 |
| Employ (1 = yes, 0 = no) | 40,770 | 0.767 | 0.423 | 0.000 | 1.000 |
| Household size (number of family members, value: 1–21) | 40,770 | 4.227 | 1.929 | 1.000 | 21.000 |
| Gender (1 = female, 0 = male) | 40,770 | 0.487 | 0.500 | 0.000 | 1.000 |
| Houseownership (1 = yes, 0 = no) | 40,770 | 0.876 | 0.330 | 0.000 | 1.000 |
| Household income (household income per capita (log)) | 40,770 | 9.435 | 1.053 | 0.000 | 15.243 |
| Married (1 = yes, 0 = no) | 40,770 | 0.864 | 0.343 | 0.000 | 1.000 |
| Religion (1 = yes, 0 = no) | 40,770 | 0.111 | 0.314 | 0.000 | 1.000 |

### 3.3. Method

This study utilized a panel regression analysis methodology using panel data from 2014 to 2018. We used the random-effect model in the panel analysis, as the fixed-effect model might not estimate time-invariant variables, such as gender and hukou register, or may cause unexpected problems [54]. While existing studies are cross-sectional studies, this study has the advantage of employing a panel analysis that allows consideration of differences between variables and time series of differences within individuals. Among the dependent variables, social trust is an ordinary least squares regression panel model and a continuous variable, and social participation and network are a semi-log regression panel model as log-type variables. The dependent variable $y_{it}$ is the social capital level of individual $i$ in year $t$. The $\alpha$ is a constant value, and the $\beta_1$ is the regression coefficient of the Gini index and $\beta_2$ is the coefficient of the control variables. The $G_{it}$ is the value of the Gini coefficient in year $t$ of the region for $i$ individuals, and the $\chi_{it}$ is the value of each control variable in year $t$ for $i$ individuals. The $u_i$ is the individual effect that does change over time, and the $e_{it}$ is the idiosyncratic error that varies over $i$ and $t$. The formula for the panel regression analysis model is as follows:

$$y_{it} = \alpha + \beta_1 G_{it} + \beta_2 \chi_{it} + u_i + e_{it}$$
$$i = 1, 2, \ldots, n, \ t = 2014 \sim 2018$$

(1)

## 4. Empirical Analysis

### 4.1. Analysis Result of Relationship between Economic Inequality and Social Capital

This study analyzed the empirical models in three steps. First, we analyzed the relationship between economic inequality and social capital. Second, we examined social capital associated with economic inequality according to social capital elements. Third, we explored the moderating effect of hukou, indicating the importance of social status in China in the association between economic inequality and social capital. In Table 2, Model 1 applied economic inequality without hukou registration, and Model 2 applied hukou without inequality. Finally, Model 3 applied both economic inequality and hukou, and focused on the results of Model 3.

The results of Model 3 show that economic inequality significantly and negatively affects social trust. In other words, the Chinese residents' social trust deteriorates in a highly economically unequal environment. Moreover, economic inequality has a significantly negative association with the social networks. In this vein, behavior to maintain a social relationship, such as giving and receiving a gift, decreases in societies with high economic inequality. However, economic inequality had an insignificant negative correlation with

social participation. Therefore, individuals in areas with high economic inequality are more likely to have low levels of social capital. When it comes to hukou, rural hukou is positively associated with social trust as a cognitive dimension of social capital. By comparison, urban hukou is positively associated with social participation and networks as a structural dimension of social capital. In sum, the social capital relationship differs according to the differences in hukou. To sum up, the H1 and H2 proposed in this paper passed the test. This study also explores the two-way causal relationship between economic inequality and social capital, and effectively mitigates the endogeneity problem using the instrumental variables approach. Table A1 shows that, whether in the benchmark regression model or in the instrumental variables model, the regression results have a high degree of consistency. Therefore, the results of this paper are robust. The robust test is described in more detail in Appendix A.

**Table 2.** Results for regression by social capital.

| | Model 1 | | | Model 2 | | | Model 3 | | |
|---|---|---|---|---|---|---|---|---|---|
| | **Trust** | **Participation** | **Network** | **Trust** | **Participation** | **Network** | **Trust** | **Participation** | **Network** |
| | **Coef (S.E.)** | **Coef (S.E.)** | **Coef (S.E.)** | **Coef (S.E.)** | **Coef (S.E.)** | **Coef (S.E.)** | **Coef (S.E.)** | **Coef (S.E.)** | **Coef (S.E.)** |
| Economic inequality | −0.607 * | −0.776 ** | −1.413 *** | | | | −0.725 *** | −0.367 | −1.349 *** |
| | (0.335) | (0.338) | (0.287) | | | | (0.336) | (0.339) | (0.288) |
| Hukou | — | | | −0.143 *** | 0.419 *** | 0.0988 *** | −0.152 *** | 0.425 *** | 0.0742 *** |
| | | | | (0.034) | (0.032) | (0.028) | (0.035) | (0.033) | (0.029) |
| Gender | 0.192 *** | −0.0582 ** | −0.0110 | 0.196 *** | −0.0696 *** | −0.00624 | 0.195 *** | −0.0673 *** | −0.0125 |
| | (0.028) | (0.025) | (0.022) | (0.027) | (0.024) | (0.022) | (0.028) | (0.025) | (0.022) |
| Age | 0.009 *** | 0.001 | −0.0097 *** | 0.0093 *** | −0.0000 | −0.0099 *** | 0.0095 *** | −0.0004 | −0.0098 *** |
| | (0.001) | (0.001) | (0.001) | (0.001) | (0.001) | (0.001) | (0.001) | (0.001) | (0.001) |
| Education | 0.178 *** | 0.638 *** | 0.0952 *** | 0.212 *** | 0.521 *** | 0.0582 ** | 0.220 *** | 0.525 *** | 0.0752 *** |
| | (0.034) | (0.031) | (0.028) | (0.035) | (0.032) | (0.028) | (0.035) | (0.032) | (0.029) |
| Married | 0.0225 | 0.144 *** | 0.340 *** | 0.0418 | 0.113 *** | 0.339 *** | 0.0273 | 0.129 *** | 0.337 *** |
| | (0.038) | (0.036) | (0.031) | (0.037) | (0.035) | (0.031) | (0.038) | (0.036) | (0.031) |
| Employ | 0.0320 | 0.0732 *** | 0.0755 *** | 0.00707 | 0.133 *** | 0.0777 *** | 0.0151 | 0.126 *** | 0.0843 *** |
| | (0.028) | (0.028) | (0.024) | (0.028) | (0.027) | (0.024) | (0.028) | (0.028) | (0.024) |
| Religion | −0.0525 * | 0.0963 *** | 0.136 *** | −0.0386 | 0.0902 *** | 0.157 *** | −0.0514 | 0.0942 *** | 0.136 *** |
| | (0.032) | (0.033) | (0.027) | (0.030) | (0.031) | (0.027) | (0.032) | (0.033) | (0.027) |
| Household income | 0.0199 * | 0.309 *** | 0.250 *** | 0.0252 ** | 0.283 *** | 0.244 *** | 0.0277 ** | 0.284 *** | 0.246 *** |
| | (0.011) | (0.011) | (0.009) | (0.011) | (0.011) | (0.009) | (0.011) | (0.011) | (0.010) |
| Household size | −0.00451 | 0.0565 *** | 0.0760 *** | −0.00742 | 0.0590 *** | 0.0738 *** | −0.00534 | 0.0583 *** | 0.0764 *** |
| | (0.007) | (0.006) | (0.006) | (0.007) | (0.006) | (0.005) | (0.007) | (0.006) | (0.006) |
| Apartment | −0.249 *** | 0.533 *** | 0.0430 | −0.187 *** | 0.377 *** | 0.0368 | −0.199 *** | 0.378 *** | 0.0172 |
| | (0.032) | (0.032) | (0.027) | (0.033) | (0.033) | (0.028) | (0.034) | (0.034) | (0.029) |
| Houseownership | 0.0918 *** | −0.0250 | 0.292 *** | 0.0950 *** | −0.0202 | 0.294 *** | 0.0892 *** | −0.0169 | 0.293 *** |
| | (0.033) | (0.033) | (0.028) | (0.032) | (0.032) | (0.028) | (0.033) | (0.033) | (0.028) |
| Constant | 6.160 *** | −2.203 *** | 5.401 *** | 5.850 *** | −2.359 *** | 4.806 *** | 6.156 *** | −2.198 *** | 5.404 *** |
| | (0.202) | (0.202) | (0.172) | (0.127) | (0.126) | (0.109) | (0.202) | (0.202) | (0.172) |
| Observations | 40,770 | 40,770 | 40,770 | 42,612 | 42,612 | 42,612 | 40,770 | 40,770 | 40,770 |
| Wald chi2 | 231.40 | 2460.26 | 1464.37 | 245.61 | 2672.69 | 1484.06 | 199.97 | 2640.06 | 1470.84 |
| Prob > chi2 | <0.0001 | <0.0001 | <0.0001 | <0.0001 | <0.0001 | <0.0001 | <0.0001 | <0.0001 | <0.0001 |

Note: Standard errors are in parentheses. ***, **, and * denote significance at the 1%, 5%, and 10% levels, respectively.

### 4.2. Moderating Effect of Hukou

Models 1 (social trust), 2 (social participation), and 3 (social network) in Table 3 included the interaction variables of hukou with economic inequality and explored social capital by cognitive and structural dimensions, respectively. Among the social capital elements, social trust and participation show a moderating effect of hukou on economic inequality. Regarding social trust (Model 1), hukou shows a moderating effect on the association between social trust and economic inequality ($B = 1.906$, $p = 0.001$). Specifically, Figure 1 illustrates that rural hukou have relatively lower social trust than urban hukou in an environment of high economic inequality. Regarding social participation (Model 2), like social trust, hukou has a moderating effect on the correlation between social participation and economic inequality ($B = 2.509$, $p = 0.001$). Figure 2 also shows that rural hukou have a more decreasing social participation behavior than urban hukou in an environment of high economic inequality. In other words, the moderating effects suggest that rural hukou are more disadvantageous to social capital formation than urban hukou in an environment of high economic inequality. However, regarding social networks (Model 3),



hukou's moderating effect is insignificant in the social networks ($B = 0.227$, $p = 0.001$). As shown in Figure 3, hukou does not show a significant moderating effect on the association between social networks and economic inequality. In summary, the results indicate that the negative association between economic inequality and social capital differs depending on the hukou type. According to the above empirical analysis, H2 and H3 are all satisfied, and the negative association between inequality and social capital is more unfavorable for individuals with rural hukou registers, and it also differs according to the social capital element.

**Table 3.** Moderation effect of social capital.

| | Model 1 | Model 2 | Model 3 |
|---|---|---|---|
| | Social Trust | Social Participation | Social Network |
| | Coef (S.E.) | Coef (S.E.) | Coef (S.E.) |
| Economic inequality ⓖ | −1.302 *** (0.400) | −1.138 *** (0.406) | −1.419 *** (0.343) |
| Hukou ⓗ | −1.023 *** (0.330) | −0.721 ** (0.332) | −0.0295 (0.282) |
| Moderation ⓖ*ⓗ | 1.906 *** (0.718) | 2.509 *** (0.723) | 0.227 (0.614) |
| Gender | 0.194 *** (0.028) | −0.0680 *** (0.025) | −0.0126 (0.022) |
| Age | 0.0095 *** (0.001) | −0.0004 (0.001) | −0.0098 *** (0.001) |
| Education | 0.219 *** (0.035) | 0.523 *** (0.032) | 0.0750 *** (0.029) |
| Married | 0.0265 (0.038) | 0.128 *** (0.036) | 0.337 *** (0.031) |
| Employ | 0.0142 (0.028) | 0.125 *** (0.028) | 0.0842 *** (0.024) |
| Religion | −0.0515 (0.032) | 0.0938 *** (0.033) | 0.136 *** (0.027) |
| Household income | 0.0279 ** (0.011) | 0.284 *** (0.011) | 0.246 *** (0.010) |
| Household size | −0.00557 (0.007) | 0.0580 *** (0.006) | 0.0764 *** (0.006) |
| Apartment | −0.196 *** (0.034) | 0.384 *** (0.034) | 0.0175 (0.029) |
| Houseownership | 0.0889 *** (0.033) | −0.0174 (0.033) | 0.294 *** (0.028) |
| Constant | 6.425 *** (0.226) | −1.838 *** (0.227) | 5.435 *** (0.193) |
| Observations | 40,770 | 40,770 | 40,770 |
| Wald chi2 | 207.34 | 2654.34 | 1472.42 |
| Prob >chi2 | <0.0001 | <0.0001 | <0.0001 |

Note: Standard errors are in parentheses. And ***, ** denotes significance at the 1%, 5% level, respectively.

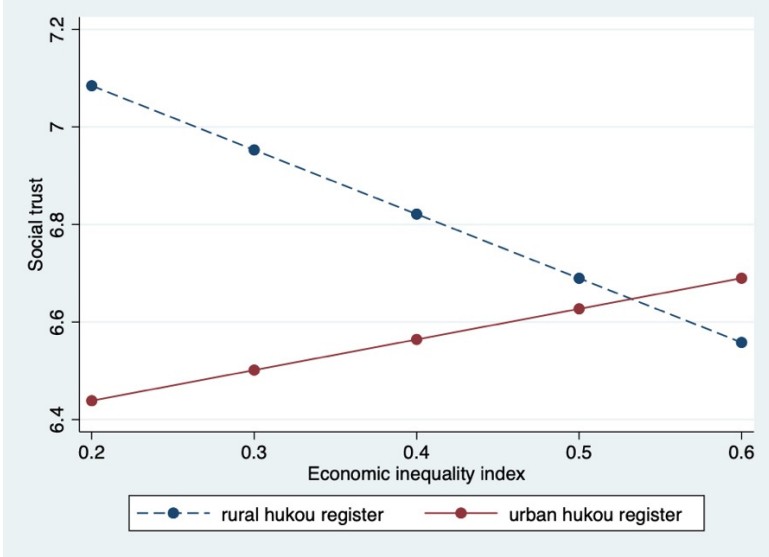

**Figure 1.** Moderation effect of social trust.

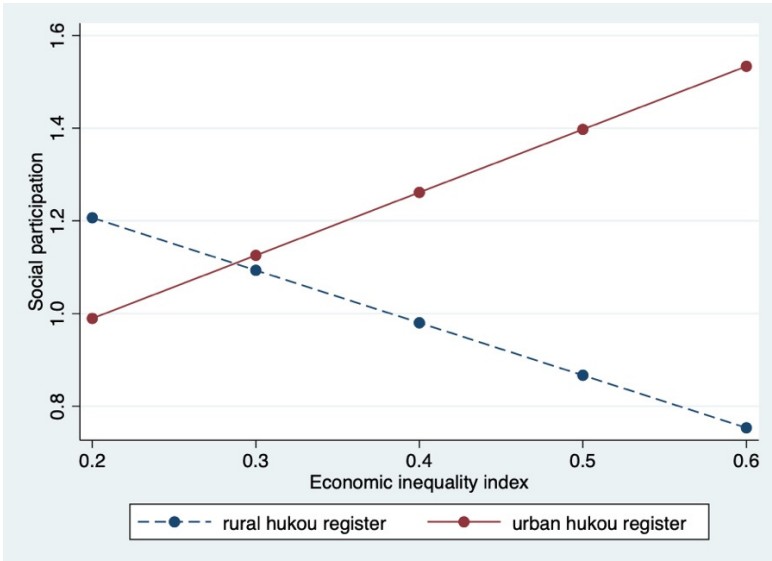

**Figure 2.** Moderation effect of social participation.

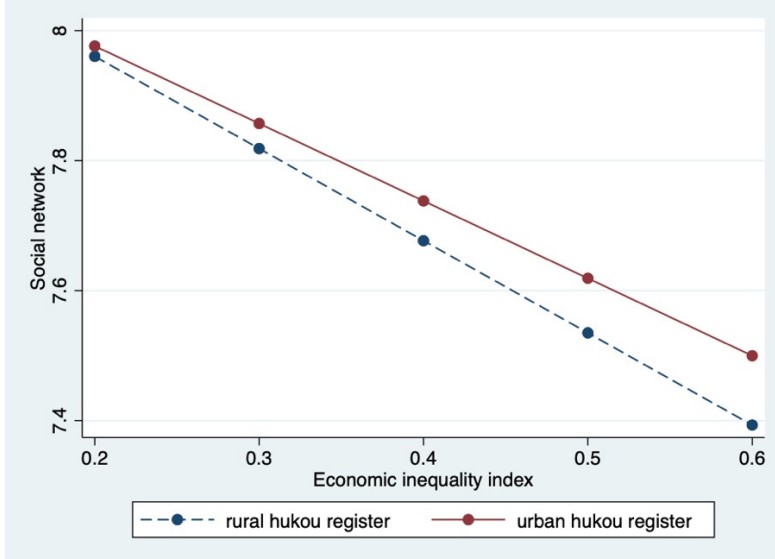

**Figure 3.** Moderation effect of the social network.

## 5. Discussion and Conclusions

To verify the assumption that Chinese residents would reduce the social capital in society due to severe economic inequality, an analysis is conducted in this study as to the correlation between the economic inequality index and social capital perception. The data sourced from the China Family Panel Studies are used to measure the economic inequality index from 2014 to 2018 for 25 Chinese provinces, municipalities, and autonomous regions, where the continuous worsening of economic inequality is an emerging social issue. The analytical models consider the social capital of cognitive elements, such as the social trust in neighborhood, and various structural elements, such as social participation and networks. An analysis is carried out to explore the association between the economic inequality index and social capital in a model divided by social capital elements. Then, the core variable of hukou is introduced to verify whether hukou moderates the relationship between economic inequality and social capital.

First, the level of individual social capital is low in those areas with a high economic inequality index. That is to say, the level of social capital would decrease due to a severe economic inequality environment. Second, there are variations in the negative association

between economic inequality and the level of social capital, which depends on those social capital factors. The negative association reaches a significant extent for social trust and networks, rather than social participation. Third, hukou plays a role in moderating the correlation between economic inequality and social capital. This negative association is more pronounced for rural hukou than for urban hukou. That is to say, rural hukou have undesirably reduced social capital compared to urban hukou. In other words, rural hukou is disadvantaged in an area with severe economic inequality, even in terms of social capital.

The first finding of this study is a negative association between economic inequality and social capital, as discovered in China. In this study, it was revealed that the level of individual social trust and networks between neighbors is low in the regions with a high economic inequality index. In other words, the mechanism through which individuals' social capital decreases in the regions with significant economic disparities and this also applies in China. The results of this study are essentially consistent with those of previous studies conducted in Western countries [1,11,25,27]. It means that the social mechanism followed in Western society is applicable in China, despite the difference in politics and cultures. In addition, these results support the study of Dai et al. (2020) [55], who studied the link between wealth inequality and social capital, and that of He et al. (2021) [56], who explored the association between social capital and Kawai index. This is a relative individual difference in China. According to the results of this study, China also needs to recognize that interpersonal social capital would decrease if economic inequality increased.

In this study, various social capital elements are applied to determine the association between economic inequality and social capital. Based on the cognitive and structural aspects emphasized in the study by Harpham et al. (2002) [26], and that of Subramanian et al. (2002) [57], this study takes into consideration social trust, social participation, and social networks as social capital elements. The results of this study show that social trust and social networks are the social capital elements consistently showing a negative association with economic inequality, while social participation shows no statistical significance. In other words, the level of social trust and social networks with neighbors is low in the regions with severe economic inequality. According to the results of the questionnaire survey conducted for this study, social trust and the act of reciprocal gifting to neighbors are reduced in the area with a high economic inequality index. The finding of low social trust and social networks in an unequal environment supports the results of previous studies [27,57]. By contrast, it is a fact that the low level of social participation is not statistically significant, even in an unequal environment. That is to say, participation characteristics differ from trust and reciprocal networks. While gifting and social trust form a social capital that the individual's volition can be readily changed, social participation, like collaboration, is operated by the participants' volition [58] and requires a prevenient interaction [59,60]. Unlike social trust and reciprocal networks according to individual volition, social participation is a collective characteristic that has a collective nature. Therefore, it is predicted that the changes in social participation are insignificant, even in case of severe economic inequality.

In an unequal society, social status is considered more significant, and low social status causes psychological withering [13]. If this concept applies to China, it means that hukou registration, which indicates social status by the place of birth, becomes more prominent and important in an unequal society. In a society with socioeconomical inequality, the importance of hukou registers, which represents social status, is highlighted. Additionally, there is an evident inequality in rural hukou registers. These results are attributed to the social structure in which the right to education and mobility of rural hukou are more restricted than urban hukou. Further, their performance in income and education [16] and access to the beneficial resources are reduced [18]. Therefore, it is concluded that, in an unequal society, the restrictions on performance and access have a negative impact on the social trust and reciprocal behavior of rural hukou families. This result also illustrates that the hukou registration system causes inequality in social capital, as analyzed in considerable research efforts on social capital [19,20,61,62].

This study is subjected to several limitations. There is no analysis of social capital by dividing it into heterogeneous and homogeneous social capital, as emphasized in the study by Lin (2002) [21]. In this study, the focus is placed on verifying the assumption that heterogeneous social capital decreases in regions with severe economic inequality. According to the results of this study, the negative association with economic inequality is attributable to heterogeneous social capital, rather than homogeneous social capital. However, since the analytical data contribute nothing to distinguishing whether social capital is a homogeneous exchange or a heterogeneous one, this analysis plays a limited role in distinguishing between these types of social capital. In addition, the model of this study is incapable of categorizing regions into urban and rural areas. The results of the moderating effect of family registration are more likely to manifest themselves in urban areas than in rural areas. In other words, it is likely to encounter difficulty in gaining social capital for those rural hukou families migrating to cities rather than those staying in rural areas. However, there is no way to separate and only analyze urban residents in this study due to the limitations in data analysis. Therefore, it is necessary to divide and analyze urban and rural areas in future research. Moreover, in the section of endogeneity treatment, despite the control applied on as many variables affecting social capital as possible, the empirical findings are still influenced by the endogeneity problem in the model. Endogeneity may also result from the two-way causality arising from the interaction between income inequality and social capital. Herein, there are limitations in how to address the endogeneity problem that cannot be fully considered for analysis. For this reason, a more in-depth analysis of endogeneity is required in future research.

**Author Contributions:** Conceptualization, S.L., S.K. and J.-H.K.; methodology, S.L. and S.K.; software, S.L.; validation, S.L., S.K. and J.-H.K.; formal analysis, S.L.; investigation, S.L. and S.K.; resources, S.L.; data curation, S.L.; writing—original draft preparation, S.L. and S.K.; writing—review and editing, S.L. and S.K.; visualization, S.L., S.K. and J.-H.K.; supervision, J.-H.K.; project administration, J.-H.K. All authors have read and agreed to the published version of the manuscript.

**Funding:** This study received no external funding.

**Institutional Review Board Statement:** Not applicable.

**Informed Consent Statement:** Not applicable.

**Data Availability Statement:** The dataset generated and analysed during the current study is not publicly available but is available from the corresponding author on reasonable request.

**Conflicts of Interest:** The authors declare no conflict of interest.

## Appendix A

Two-way causation is another factor contributing to endogeneity, in addition to missing control variables. Economic inequality and social capital have a causal relationship in this study, meaning that social capital will change as a result of economic inequality. In order to avoid the estimation bias caused by the endogeneity problem and to test the robustness of the results of the benchmark regression analysis., this study adopts the IV-Three stage least squares estimation method of instrumental variables for analysis. The 2012 CFPS Gini index is used in this study as an instrumental variable for measuring economic inequality. There is no weak instrumental variable issue in the instrumental variable test because the statistic's *p*-value is less than 0.0001. Table A1 displays the results of the regression using the IV-3SLS method following the addition of instrumental variables. We also concentrate on social trust and social networks in this context because social participation is not statistically significant in the benchmark regression results. Comparing the baseline regression results, whether in the benchmark regression model or in the IV model, the regression results have a high degree of consistency, and the results of this paper are robust. Economic inequality is indeed an important reason for inhibiting residents' social capital level, and the expansion of economic inequality will reduce the level of social capital of the rural residents.

**Table A1.** IV-Three stage least squares regression results.

| | Model 1 | Model 2 | Model 3 |
|---|---|---|---|
| | **Social Trust** | **Social Participation** | **Social Network** |
| | **Coef (S.E.)** | **Coef (S.E.)** | **Coef (S.E.)** |
| Economic inequality (IV) | −8.136 *** (1.505) | 1.806 (1.455) | −14.45 *** (1.259) |
| Hukou | −0.190 *** (0.040) | 0.489 *** (0.038) | −0.0432 (0.033) |
| Gender | 0.191 *** (0.026) | −0.0638 ** (0.025) | −0.0184 (0.022) |
| Age | 0.00838 *** (0.001) | −0.000318 (0.001) | 0.00922 *** (0.001) |
| Education | 0.291 *** (0.035) | 0.534 *** (0.034) | 0.102 *** (0.029) |
| Married | 0.0389 (0.041) | 0.127 *** (0.039) | 0.351 *** (0.034) |
| Employ | 0.0301 (0.034) | 0.175 *** (0.033) | 0.133 *** (0.028) |
| Religion | −0.0843 ** (0.041) | 0.169 *** (0.040) | 0.0311 (0.035) |
| Household income | 0.0559 *** (0.014) | 0.294 *** (0.013) | 0.313 *** (0.011) |
| Household size | 0.000679 (0.007) | 0.0597 *** (0.007) | 0.0906 *** (0.006) |
| Apartment | −0.319 *** (0.041) | 0.435 *** (0.040) | −0.126 *** (0.035) |
| Houseownership | 0.0917 ** (0.043) | 0.00520 (0.042) | 0.293 *** (0.036) |
| Constant | 9.333 *** (0.700) | −3.408 *** (0.676) | 10.76 *** (0.585) |
| Observations | 28,257 | 28,257 | 28,257 |
| Wald chi2 | 287.63 | 2695.79 | 1576.34 |
| Prob > chi2 | <0.0001 | <0.0001 | <0.0001 |

Note: Standard errors are in parentheses. And ***, ** denotes significance at the 1%, 5% level, respectively.

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
