# Peer review of "Social Capital Inequality According to Hukou in Unequal Economic Environments in China"

_sustainability, doi:10.3390/su142113748_

Round 1
Reviewer 1 Report
This manuscript explored social capital inequality according to Hukou in unequal economic environments in China, using panel regression analysis for methodology.
1. There is an error in the institution of affiliation for some of the authors: Hangyang University or Hanyang University (page 1, lines 6-7)
2. I propose to the authors insert a paragraph to explain the keyword “Hukou” before presenting rural hukou and urban hukou (Section 1. Introduction, page 1 or 2).
3. This study draws attention to a topic of interest in the field of hukou migration. I also recommend the authors' papers to correlate the work related to measuring effects (Tan et al, 2021 - https://doi.org/10.1080/1331677X.2021.2010110)
4. I could not identify in the content of the manuscripts, the validation of the hypotheses. The authors summarize the state of the hypotheses in the Abstract Section and carry out their definition in section 2.3. Contribution (H1-H3). Subsequently, they are no longer present (page 3, 138-143).
5. In section 4.2. Moderating effect of hukou. Why did you opt not to show the value for the hukou's moderating effect in the social network, even if the effect is insignificant? Authors can insert form Model 3 – B 0,227, p 0,001 – page 8, line 294-295). Also, authors can make a new graphic representation for model 3 like Figure 3 - Moderation effect of the social network (page 9, line 306.
Author Response
Reviewer 1
Dear Reviewer:
We are grateful to Reviewer for reviewing the paper so carefully. Thank you for your decision and constructive comments on my manuscript. We have tried our best to improve and made some changes to the manuscript. The red part has been revised according to your comments. Revision notes, point-to-point, are given as follows:
- There is an error in the institution of affiliation for some of the authors: Hangyang University or Hanyang University (page 1, lines 6-7)
In agreement with your advice, we have revised and supplemented the manuscript as follows. “Supplementary Content.” (Page 1, lines 6-7)
Department of Urban and Regional Development, Hanyang University; namugnel@gmail.com
Department of Urban and Regional Development, Hanyang University; jhkoo@hanyang.ac.kr
- I propose to the authors insert a paragraph to explain the keyword “Hukou” before presenting rural hukou and urban hukou (Section 1. Introduction, page 1 or 2).
In agreement with your advice, we have revised and supplemented the manuscript as follows. “Supplementary Content.” (Page 2, lines 58-68)
The hukou registration system is a basic institutional arrangement in Chinese society, and it is also a system centered on hukou registration and management. [14] The hukou system takes hukou as an important credential for resource allocation and benefit distribution, which has a great impact on social stratification and mobility [14,15]. The core content of the hukou registration system includes a dual identity system that divides citizens into urban and rural registrations: at the same time, according to the principle of hukou jurisdiction, strict administrative control is exercised on the migration of hukou between different places. This institutional arrangement has had an important impact on the formation of the urban-rural dual structure of Chinese society and the emergence of urban hierarchies through the control of identity conversion and autonomous migration [14].
- This study draws attention to a topic of interest in the field of hukou migration. I also recommend the authors' papers to correlate the work related to measuring effects (Tan et al, 2021 - https://doi.org/10.1080/1331677X.2021.2010110)
In agreement with your advice, we have revised and supplemented the manuscript as follows. “Supplementary Content.” (Page 2, lines 60-62)
The hukou system takes hukou as an important credential for resource allocation and benefit distribution, which has a great impact on social stratification and mobility [15].
- I could not identify in the content of the manuscripts, the validation of the hypotheses. The authors summarize the state of the hypotheses in the Abstract Section and carry out their definition in section 2.3. Contribution (H1-H3). Subsequently, they are no longer present (page 3, 138-143).
In agreement with your advice, we have revised and supplemented the manuscript as follows. “Supplementary Content.” (Page 7-8, lines 298-299, 321-324)
To sum up, H1 and H2 proposed in this paper pass the test.
According to the above empirical analysis, H2 and H3 are all satisfied, and the negative association between inequality and social capital is more unfavorable for individuals with rural hukou registers, and it also differs according to the social capital element.
- In section 4.2. Moderating effect of hukou. Why did you opt not to show the value for the hukou's moderating effect in the social network, even if the effect is insignificant? Authors can insert form Model 3 – B 0,227, p 0,001 – page 8, line 294-295). Also, authors can make a new graphic representation for model 3 like Figure 3 - Moderation effect of the social network (page 9, line 306.
In agreement with your advice, we have revised and supplemented the manuscript as follows. “Supplementary Content.” (Page 8,10 lines 316-319,333-334 )
On the other hand, regarding social networks (Model 3), hukou's moderating effect is insignificant in the social networks (B = 0.227, p = 0.001). As shown in Figure 3, hukou does not show a significant moderating effect on the association between social networks and economic inequality.
There is a figure in word file.

Reviewer 2 Report
Sustainability - 1923221
The study deals with an interesting topic that of the relationship between social capital stock at the regional level and economic inequality. The data are appropriate and the use of a registry to account for level of urbanization is also important. However in its current form the study needs major revisions to provide:
a) A clear, concise and informative discussion of the interrelationships between social capital stock and economic inequality and urbanization, and most importantly discuss the endogeneity effects among them. This discussion should then inform the empirical tests of the study. It is essential that the authors test for two-way causation between social capital and economic inequality and inform their empirical estimations, results discussion etc. according to the endogeneity tests.
b) A second major concern relates to the confusing presentation of the text. Extensive syntax changes are needed for the text to clearly present its argument and potential contribution. These changes should be made along the lines of accommodating comment 1 above (e.g. there is a confusing presentation of the discussion between variations in social capital and economic inequality).
c) At the empirical level you need to add a paragraph wherein to present the Hukou registry, its methodology, data etc. While the study uses data that are made available via the Hukou registry I would suggest that reference to Hukou (e.g. in the title, in section 2.2. etc.), is made more clear.
Author Response
Reviewer 2
Dear Reviewer:
We are grateful to Reviewer for reviewing the paper so carefully. Thank you for your decision and constructive comments on my manuscript. We have tried our best to improve and made some changes to the manuscript. The red part has been revised according to your comments. Revision notes, point-to-point, are given as follows:
- a) A clear, concise and informative discussion of the interrelationships between social capital stock and economic inequality and urbanization, and most importantly discuss the endogeneity effects among them. This discussion should then inform the empirical tests of the study. It is essential that the authors test for two-way causation between social capital and economic inequality and inform their empirical estimations, results discussion etc. according to the endogeneity tests.
We do agree with you that there is an endogeneity problem in this study. However, due to the limitations of the endogeneity problem analysis, we are afraid that we have to explore the endogeneity problems as part of the discussion and analyzed and address the endogeneity problem in detail in future studies. In agreement with your advice, we have revised and supplemented the manuscript as follows. “Supplementary Content.” (Page 12 lines 421-427 )
Moreover, in the endogeneity treatment section, although this paper controls for as many variables affecting social capital as possible, the empirical findings are still threatened by the endogeneity problem in the model. Endogeneity may also arise because of the two-way causality that occurs as a result of the interaction between income inequality and social capital. In this study, there are limitations to the endogeneity problem that cannot be fully considered for analysis. A more in depth and detailed analysis of the endogeneity issue is necessary for future research.
- b) A second major concern relates to the confusing presentation of the text. Extensive syntax changes are needed for the text to clearly present its argument and potential contribution. These changes should be made along the lines of accommodating comment 1 above (e.g. there is a confusing presentation of the discussion between variations in social capital and economic inequality).
We apologize for the poor language of our manuscript. We worked on the manuscript for a long time and the repeated addition and removal of sentences and sections obviously led to poor readability. We have now worked on both language and readability and have also involved native English speakers in language corrections. We really hope that the flow and language level have been substantially improved.
In agreement with your advice, we have made numerous grammatical changes and touches to the manuscript to present the arguments and potential contributions more clearly and smoothly. We have revised and supplemented the manuscript as follows. “Supplementary Content.” (Page 1-12)
Please see the uploaded attachment for details of the grammatical changes.
- c) At the empirical level you need to add a paragraph wherein to present the Hukou registry, its methodology, data etc. While the study uses data that are made available via the Hukou registry I would suggest that reference to Hukou (e.g. in the title, in section 2.2. etc.), is made more clear.
In agreement with your advice, we have revised and supplemented the manuscript as follows. “Supplementary Content.” (Page 2,5 lines 58-68, 218,232-238)
The hukou registration system is a basic institutional arrangement in Chinese society, and it is also a system centered on hukou registration and management. [14] The hukou system takes hukou as an important credential for resource allocation and benefit distribution, which has a great impact on social stratification and mobility [14,15]. The core content of the hukou registration system includes a dual identity system that divides citizens into urban and rural registrations: at the same time, according to the principle of hukou jurisdiction, strict administrative control is exercised on the migration of hukou between different places. This institutional arrangement has had an important impact on the formation of the urban-rural dual structure of Chinese society and the emergence of urban hierarchies through the control of identity conversion and autonomous migration [14].
Each citizen is required to legally register at a household police station from birth, and this registration is known as personal identification. (hung 2022) The household registration records the type of hukou type, legal address, up to affiliation, and the other personal and family details of Chinese citizens(wang 2005). In this paper, we use the question, “current household registration type is_?” The type of “Agricultural” is rural hukou, and the type of “Non-Agricultural” is urban hukou. So in this paper, hukou, a dummy variable, is coded as rural hukou =0 and urban hukou=1.
Reviewer 3 Report
In Abstract more precise formulate aim and better description of research methods applied in the paper as the style used in this paper could be better in description of applied research methods.
It is recommended in table 1 (Variable list and descriptive statistics) add also mode and median as well as after the table give at least some explanation of information included in the table. It is recommended to state more precise - it is not understandable what mean “Household size(reference: 1 person)” and “Household income (reference: 1 person)”.
In formula 1 state more precise regression coefficients ? as the results of empirical analysis show that it is not the same value, but in the formula there are indicated ? for two different variables.
Not indicated pubishing year “Rummel, N.; Deiglmayr, A.; Spada, H.; Kahrimanis, G.; Avouris, N. Analyzing collaborative interactions across domains and 500
settings: An adaptable rating scheme. In Analyzing interactions in CSCL (pp. 367-390). Springer, Boston, MA”
Author Response
Reviewer 3
Dear Reviewer:
We are grateful to Reviewer for reviewing the paper so carefully. Thank you for your decision and constructive comments on my manuscript. We have tried our best to improve and made some changes to the manuscript. The red part has been revised according to your comments. Revision notes, point-to-point, are given as follows:
1. In Abstract more precise formulate aim and better description of research methods applied in the paper as the style used in this paper could be better in description of applied research methods.
In agreement with your advice, we have revised and supplemented the manuscript as follows. “This study aims to explore a social phenomenon in which individual social capital decreased in economically unequal regions in China (Page 1, lines 10-11). Since these data are panel data surveyed from 2014 to 2018, this study used the random-effect model in the panel analysis for methodology (Page 1, lines 13-15).”
- It is recommended in table 1 (Variable list and descriptive statistics) add also mode and median as well as after the table give at least some explanation of information included in the table. It is recommended to state more precise - it is not understandable what mean “Household size(reference: 1 person)” and “Household income (reference: 1 person)”.
We have tried to include the mode and median in Table 1, but after test and experimenting with finding the prior study, we believe that there is no significant difference in these values, so for the sake of clarity we are afraid to use Mean as a basis for this study. And in agreement with your advice, we state more precisely the definition of the variable list and descriptive statistics, and we have revised and supplemented the manuscript as follows. “Supplementary Content” (Page 6, lines 257-258)
3. In formula 1 state more precise regression coefficients ? as the results of empirical analysis show that it is not the same value, but in the formula there are indicated ? for two different variables.
In agreement with your advice, we have revised and supplemented the manuscript as follows. “Supplementary Content.” (Page 6,7 lines 269-270, 275-276)
and the β_1 is the regression coefficient of the Gini index and β_2 is the coefficient of the control variables.
y_it=α+〖〖β_1 G〗_it+β_2 χ〗_it+u_i+e_it
i = 1, 2, …, n, t = 2014 ~ 2018 (1)
4. Not indicated pubishing year “Rummel, N.; Deiglmayr, A.; Spada, H.; Kahrimanis, G.; Avouris, N. Analyzing collaborative interactions across domains and 500
settings: An adaptable rating scheme. In Analyzing interactions in CSCL (pp. 367-390). Springer, Boston, MA”
In agreement with your advice, we have revised and supplemented the manuscript as follows. “Supplementary Content.” (Page 14 lines 535-536)
Rummel, N.; Deiglmayr, A.; Spada, H.; Kahrimanis, G.; Avouris, N. Analyzing collaborative interactions across domains and settings: An adaptable rating scheme. In Analyzing interactions in CSCL. Springer, Boston, MA. 2011, 367-390.

Round 2
Reviewer 2 Report
I have seen the revised version of the manuscript and the authors' effort to incorporate the suggested revisions. Many parts of the study have been improved yet there remain the major drawbacks that were mentioned in my initial review of the manuscript. In particular, the relationship between social capital and economic inequality is central in the analysis and thus the potential effects from two-way causality cannot be just mentioned as a simple limitation of the study. It is a well-established fact that social capital affects growth so when the two variables are analyzed together we may not disregard this fact and its effect in the estimated parameters.
A second major issue relates to the structure of arguments as presented in the various parts of the study and in particular the introduction and theoretical parts.
A third issue relates to the clarity of the presented arguments and discussion in the text. The use of an English language editor has much benefited the text. However, while in some parts the syntax is correct, the meaning remains vague (e.g. 'This study aims to explore a social phenomenon in which individual social capital decreased in economically unequal regions in China'). I would suggest that the authors consistently use notions throughout the text so as for the meanings and arguments to be clear, e.g. social capital variations in developed vs deprived regions of the country.
Overall the manuscript needs to go through the initial revisions again and rewrite and restructure the arguments presented in it.
Author Response
.

Round 3
Reviewer 2 Report
The authors have tried to accommodate the suggested revisions.
Author Response
Response to comment:
The author have tried to accommodate the suggested revisions.
Response: We sincerely thank you for carefully reviewing this manuscript, your decision, and your constructive comments. Your comments have greatly helped us improve the quality of the manuscript. And we also appreciate your recognition and encouragement to us.
Once again, thank you very much for your comments and suggestions.
Kind regards,